# Media Exposure and Vicarious Trauma: Italian Adaptation and Validation of the Media Vicarious Traumatization Scale and Its Impact on Young Adults’ Mental Health in Relation to Contemporary Armed Conflicts

**DOI:** 10.3390/ejihpe15090184

**Published:** 2025-09-12

**Authors:** Giorgio Maria Regnoli, Gioia Tiano, Barbara De Rosa

**Affiliations:** Department of Humanities, University of Naples Federico II, Via Porta di Massa 1, 80133 Naples, Italy; giorgiomaria.regnoli@unina.it (G.M.R.); gio.tiano@studenti.unina.it (G.T.)

**Keywords:** media vicarious traumatization, media exposure, mental health, war, young adults

## Abstract

In recent years, psychological research has increasingly focused on the impact of media exposure on mental health, identifying young adults as particularly vulnerable due to their high levels of media engagement. To explore these effects, the construct of Media Vicarious Traumatization (MVT) has been introduced as an extension of vicarious traumatization, aimed at capturing the psychological impact of emotionally intense media content. MVT offers a relevant framework for understanding the mental health risks of media exposure, especially in relation to socially significant issues like war, now central in contemporary media discourse. This study aims to culturally adapt and psychometrically validate the Media Vicarious Traumatization Scale (MVTS) within the Italian context, and to investigate the relationship between the war-related MVT construct, generalized anxiety, and future anxiety among young adults. Study I, conducted on a sample of 250 participants (M = 22.40, SD = 2.63), explored the latent structure of the MVTS using Parallel Analysis and Exploratory Factor Analysis (EFA), yielding promising psychometric properties in terms of reliability and factorial stability. An independent sample of 553 participants (M = 22.43, SD = 2.37) was recruited for Study II to confirm the MVTS’s latent structure via Confirmatory Factor Analysis (CFA), which indicated good model fit. This study also evaluated measurement invariance across gender, internal consistency, and convergent, discriminant, and predictive validity, alongside psychometric properties assessed through Item Response Theory (IRT). The results of both studies confirm the stable and robust psychometric properties of the scale. Furthermore, Study II provides novel insights into the predictive role played not only by the war-related MVT but also by the recently introduced construct of Worry about War in exacerbating both generalized anxiety and future anxiety among Italian young adults.

## 1. Introduction

The contemporary landscape is marked by events with high traumatic potential, to which individuals—especially young people—are constantly exposed due to the exponential development of media technologies. From Russia’s invasion of Ukraine to the escalating intensity of the Israel–Hamas conflict, news coverage, images, and videos of war have acquired an invasive presence in media communication ([55]), placing these conflicts at the center of global social, political, and economic discourse ([10]; [99], [100]). In addition to affecting the mental health of populations directly involved in these conflicts ([51]; [78]; [93]), recent studies have shown that wars are also contributing to psychological distress and negative emotions in populations not directly impacted by them ([23]; [38]; [83]; [74], [72]). As highlighted by the [27] ([27]), even in countries not at war, there is a noticeable increase in public concern, uncertainty, and anxiety related to the risk of escalation, potential national involvement in military conflicts, and the possible use of chemical and nuclear weapons ([9]; [13]; [23]; [35]; [59]; [49]; [72], [70]). Wars are now pervasively present across mass media and social media, which disseminate a constant stream of news, images, and videos depicting their destructiveness and violence, with tangible repercussions for psychological well-being ([44]; [55]; [35]).

Contemporary media communication is increasingly characterized by a pathemic mode of information delivery, which, through the pervasive use of dramatic language and emotionally charged imagery, tends to elicit intense emotional responses in the audience rather than fostering a deeper understanding of events ([56]; [67]; [104]). The phenomenon of Infodemic, first described during the COVID-19 pandemic, refers to the tendency to select and disseminate chaotic and emotion-laden information, and appears to typify the current media communication landscape ([80]). It was precisely the collective trauma of the pandemic ([43]; [26]) that underscored how continuous media exposure to distressing content contributed to the onset of anxious and post-traumatic symptoms ([34]; [57]), especially among individuals with a heightened disposition toward anxiety, worry, and negative future outlooks ([62]; [103]; [69], [77], [75], [71]; [64]).

Much like the pandemic, war today constitutes a global critical event and a macro-social stressor ([15]; [74]) that heightens the perceived threat to one’s own life and the lives of others, as well as uncertainty regarding the present and future ([24]; [72]). According to Media Dependency Theory ([8]), it is precisely during times of crisis, conflict, and major societal change that individuals, in an effort to reduce uncertainty, increase their information-seeking behaviors—along with the risk of developing media dependency. Naturally, seeking information serves several adaptive functions: enhancing understanding of events (social understanding function), supporting the formation of personal opinions (personal orientation function), and guiding behavioral choices (action orientation function). However, during periods of crisis—whether political, economic, social, environmental, or health-related—the likelihood of developing a dysfunctional dependency on media increases, thereby elevating the risk of distorted perceptions of reality and, more broadly, contributing to negative psychological outcomes due to excessive exposure to distressing media content ([8]).

### 1.1. The Traumatizing Power of the Media: From Traditional Vicarious Trauma to Media Vicarious Traumatization

Taylor and Frazer’s multilevel trauma theory ([94]) has long emphasized that traumatic events can have negative psychological effects not only on directly affected individuals and communities, but also on those who are indirectly exposed. This includes the development of anxiety and post-traumatic symptoms, which may arise not only due to individual vulnerability and dynamics of identification with the victims, but also as a consequence of constant and pervasive media exposure to the traumatic event (levels 5 and 6) ([94]). The recent review by [80] ([80]) further confirmed that prolonged exposure to images and news about disastrous events significantly increases the likelihood of indirect traumatic effects, especially in individuals considered at risk due to age or personality traits. Multiple studies have supported these findings by exploring the psychological impact of media-mediated exposure to traumatic or potentially traumatic events, from the pandemic trauma ([54]; [80]) to the climate crisis ([1]) and, more recently, to contemporary wars ([55]; [33]). This body of evidence highlights the role of mass media and social media as potential catalysts of “indirect grief” ([68]), which consciously and unconsciously activates experiences of loss and psychological distress typical of traumatic processes and—specifically in the context of this study—of vicarious trauma ([91]).

[58] ([58]) were the first to define Vicarious Trauma (VT), or Secondary Traumatic Stress, as later conceptualized by [31] ([31]), as a process that can be triggered by repeated exposure to the traumatic experiences of others with whom one shares an empathetic connection ([58]). This process can lead to negative effects on emotions, cognition, and overall psychological functioning. Originally, the construct was developed to describe the long-term impact of exposure to trauma or to graphic details of traumatic experiences in the context of psychological counseling and support provided by mental health or emergency professionals ([58]; [66]). Distinct from burnout or countertransference ([52]), vicarious traumatization refers to a transformation in the helper’s cognitive schemas and imaginative memory, and it is the ongoing contact with narratives of suffering that serves as a potential trigger for the process ([65]). Over time, the notion of vicarious trauma has extended beyond the helping professions and has been employed to explore the psychological impact of indirect exposure to traumatic events across a variety of contexts ([88]), including those mediated by mass media and social media. Building on the conceptualization of vicarious trauma, several studies have demonstrated the adverse effects of prolonged exposure to media content involving disasters, terrorist attacks, destruction, and other forms of violence on mental health ([42]; [87]). [96] ([96]) have emphasized how such exposure may sustain a chronic cycle of psychological distress, potentially constituting a form of vicarious trauma ([97]). Post-traumatic stress symptoms have been observed following media exposure to the 9/11 terrorist attacks ([16]), the Boston Marathon bombing, which led to acute stress and long-term psychological distress among media consumers ([41]), and other collective traumatic events ([12]; [36]; [40]; [95]).

In examining the specific characteristics of media exposure in relation to vicarious traumatization, [28] ([28]) moved beyond earlier studies that focused solely on the type of media content. Building on the work of [58] ([58]), they highlighted the critical role of exposure frequency in contributing to emotional stress and vicarious traumatization. Supporting this emphasis on frequency, other studies ([39]; [53]) have drawn attention to the role played by social media recommendation systems, which algorithmically determine both the type and frequency of content exposure based on trending topics and user browsing history ([20]; [61]). In particular, [53] ([53]) confirm the influence of these recommendation systems in increasing the likelihood of vicarious traumatization, especially among younger users who are more frequently exposed and therefore more vulnerable. As previously noted, according to Media Dependency Theory ([8]), collective critical events increase individuals’ dependence on media as a means of managing uncertainty, thereby reinforcing the prominence of these topics—especially when amplified by algorithmic recommendation systems. Today, war represents one of the most salient critical events covered by traditional and social media, which significantly increases the chances of being exposed to high-impact traumatic content such as distressing information, images, and videos. This constant exposure to anxiety-inducing content has important consequences for users’ psychological well-being, contributing to a phenomenon of “desensitization,” which functions as a defensive mechanism to manage the emotional burden triggered by such content ([89]). The risk, however, lies in the gradual and cumulative habituation to violence and destruction, which may lead to a diminished perception of the severity of critical events and ultimately to their normalization ([84]).

In summary, numerous scientific studies underscore the significant impact of exposure to anxiety-inducing media content on users’ psychological distress, and they converge on the relevance of the construct of Media Vicarious Traumatization (MVT). However, to date, there is no universally accepted or comprehensive definition of MVT. Based on the existing literature, we propose defining Media Vicarious Traumatization (MVT) as an emotional and cognitive effect that may develop as a result of persistent and cumulative exposure to traumatic media content. It is characterized by distress, pessimism, intrusive thoughts, difficulties in emotional regulation, and a general sense of psychological overwhelm and helplessness.

In an era marked by profound uncertainty and shaped by collective events with high traumatic potential ([3]; [1]; [2]; [11]; [19]; [55]; [72], [73]), which are pervasively disseminated through a pathemic and dramatizing media discourse ([67]; [56]), it becomes increasingly urgent to adopt valid and reliable tools to investigate the role of MVT in psychological distress. This is particularly critical with respect to young people, whose pervasive use of social media makes them especially vulnerable to vicarious traumatization—a condition that may exacerbate psychological discomfort to such a degree that it now constitutes a true public health emergency in the field of mental health ([75]; [25]; [5]; [101]).

### 1.2. The Media Vicarious Traumatization Scale (MVTS)

The Media Vicarious Traumatization Scale (MVTS) was first employed by [54] ([54]) and represents a reformulation of the original Vicarious Trauma Scale (VTS) developed by [105] ([105]). The VTS was originally designed to assess self-reported levels of distress in legal professionals working with traumatized clients. It consists of seven items rated on a five-point Likert scale ranging from 1 (strongly disagree) to 7 (strongly agree), yielding a total score between 7 and 35, with higher scores indicating greater levels of vicarious traumatization. The instrument demonstrated promising psychometric properties, with an internal consistency coefficient (Cronbach’s alpha) of 0.88. Since its development, the VTS has been applied in various research contexts investigating vicarious traumatization among mental health professionals and students in social science training programs, consistently showing stable psychometric properties ([6]; [52]; [48]).

[54] ([54]) adapted and applied the original Vicarious Trauma Scale (VTS) to the media context to assess the vicarious traumatic impact of media exposure related to the COVID-19 pandemic, thereby developing the Media Vicarious Traumatization Scale (MVTS). The instrument comprises seven items rated on a five-point Likert scale (1 = strongly disagree; 5 = strongly agree) and provides a total score ranging from 7 to 35. The score, calculated as the sum of responses to each individual item, reflects the degree of media vicarious traumatization, with higher scores indicating greater distress experienced in relation to media-reported information.

In the original study, MVT emerged as a central construct mediating the relationship between information sources and anxiety symptoms during the COVID-19 pandemic and demonstrated an internal consistency of 0.78. More recently, [53] ([53]) adapted the scale to explore media vicarious traumatization in relation to an aviation accident in China, and the instrument exhibited an internal consistency of 0.89, a Composite Reliability of 0.91, and an Average Variance Extracted (AVE) of 0.60.

Currently, no instrument exists in Italy specifically deigned to assess Media Vicarious Traumatization (MVT), nor is there a version specifically calibrated for the phenomenon of ‘war,’ which today permeates media and social media, potentially impacting the mental health of their primary consumers, namely young people. Therefore, to address this gap, this study presents the Italian adaptation and validation of the MVTS calibrated for the war context, tested on a sample of Italian young adults aged 18 to 30.

### 1.3. Aims and Phases of the Research Design

This paper describes the process of the Italian adaptation and validation of the Media Vicarious Traumatization Scale (MVTS) and examines the impact of this construct—specifically contextualized to contemporary wars—on generalized anxiety and future anxiety among Italian young adults. The present research is structured into two distinct yet complementary studies.

To our knowledge, the MVTS has not previously undergone a comprehensive and rigorous psychometric validation process. Therefore, Study I aims to achieve the following:Describe the linguistic and cultural adaptation process of the MVTS into Italian;Explore the latent structure of the Italian version of the instrument and assess its preliminary psychometric properties in terms of reliability and factorial stability.

Study II, conducted on an independent sample, has two main objectives:To confirm the factorial structure of the MVTS identified in Study I and to assess—by integrating Classical Test Theory (CTT) with Item Response Theory (IRT)—additional psychometric properties, including measurement invariance across gender, internal consistency, item information across varying levels of the latent trait, and conditional reliability, as well as convergent, discriminant, and predictive validity;To investigate the role of war-related media vicarious traumatization (MVT) in exacerbating generalized anxiety and future anxiety among Italian young adults.

## 2. Study I

### 2.1. Materials and Methods

#### 2.1.1. Translation and Cultural Adaptation

The Italian linguistic and cultural adaptation of the MVTS was developed based on the original English version of the instrument ([54]), using the back-translation method. Initially, two native Italian speakers—a sociolinguist and a professional translator—independently translated the scale items. Their respective translations were then compared and synthesized into a single, agreed-upon version. This Italian version was subsequently back-translated into English by a native English speaker to ensure semantic equivalence with the source text. The entire process was carefully reviewed by the same experts, who finalized the Italian adaptation of the scale. The primary aim of the translation was to preserve the meaning of the original items while adopting language that is clear, natural, and commonly used in Italian. A simple and transparent syntactic structure was maintained, with deliberate avoidance of regional dialects. Given the already straightforward syntax of the English version, only minimal stylistic adjustments were necessary during the adaptation.

#### 2.1.2. Sample Size Determination

To determine the optimal sample size for Paralell Analysis (PA) and Exploratory Factor Analysis (EFA), the a priori estimation followed the recommendations outlined by [14] ([14]). Specifically, a minimum ratio of 20 participants per item was adopted. Based on this criterion, a sample size of 140 participants was considered adequate for the EFA, ensuring that the sample participating in Study I would yield stable and robust parameter estimates.

#### 2.1.3. Sample and Data Collection

The initial dataset comprised 255 participants and was subjected to an evaluation for missing values and outliers. Since no missing data were detected, the focus shifted to identifying multivariate outliers using Mahalanobis distance, which flagged five cases with significant values (*p* < 0.001) ([92]). These outliers were thoroughly reviewed and subsequently excluded to reduce their influence on further analyses.

The ultimate sample included 250 young adults, comprising 138 (55.2%) females and 112 (44.8%) males, aged between 18 and 30 years (M = 22.40, SD = 2.63). The majority of participants resided in Campania, a region in the south of Italy (230; 92%). Regarding relationship status, 138 (55.2%) individuals were single, while 112 (44.8%) reported being in a romantic relationship. In terms of educational attainment, 2 (0.8%) participants had completed lower secondary education, 191 (76.4%) had obtained a high school diploma, 37 (14.8%) held a bachelor’s degree, and 20 (1.2%) possessed a master’s degree. Regarding employment status, 134 (53.6%) were full-time students, 62 (24.8%) were working students, 35 (14.5%) were employed, and 19 (7.36%) were unemployed.

Data collection was conducted in Italy in January 2025 through a snowball sampling approach. The survey was administered online and promoted via social media channels, as well as printed advertisements displayed in communal areas within university campuses. To be eligible, respondents had to be Italian citizens, aged between 18 and 30 years, and had to provide informed consent at the outset of the questionnaire. Participants were clearly informed about the study’s purpose, the confidentiality of their responses, and the voluntary nature of their involvement.

#### 2.1.4. Data Analysis Plan

Preliminary descriptive analyses were conducted to examine the central tendency and dispersion of each item, including means, standard deviations, and variances. Data distribution was evaluated through skewness and kurtosis indices, with values ranging from −1.0 to 1.0 interpreted as evidence of approximate normality, in line with the guidelines established by [92] ([92]). To assess multivariate normality, Mardia’s tests for multivariate skewness and kurtosis were performed. The findings from these initial assessments guided the choice of the most appropriate estimator for the subsequent validation procedures. Following the principles of Classical Test Theory (CTT), preliminary analyses were carried out to examine the corrected item–total correlations and inter-item correlations, aiming to assess the extent to which individual items contributed to the measurement of a common underlying construct. Correlation coefficients falling below the threshold of 0.30 were interpreted as indicators of suboptimal item performance and considered for potential removal from the scale ([14]). Preliminary assessments prior to conducting Exploratory Factor Analysis (EFA) encompassed the Kaiser–Meyer–Olkin (KMO) measure, Bartlett’s test of sphericity, and the Measure of Sampling Adequacy (MSA). A KMO value exceeding 0.80 ([92]), a significant Bartlett’s test (*p* < 0.001), and MSA indices above 0.50 were regarded as satisfactory indicators supporting the appropriateness of the data for factor extraction. An Optimized Parallel Analysis (PA) utilizing minimum rank factor analysis was conducted to preliminarily investigate the latent structure of the Italian version of the MVTS. Then, EFA with robust maximum likelihood estimator (MLM) was implemented to corroborate the factorial structure merged by PA. The assessment of unidimensionality was further supported by examining indices such as Unidimensional Congruence (UniCo > 0.95), Item Explained Common Variance (ECV > 0.85), and the Mean of Item Residual Absolute Loadings (MIREAL < 0.30), applying their established threshold values as criteria. The factor solution was tested considering eigenvalues greater than 1.0, item communalities above 0.40, and standardized factor loadings greater than 0.60 ([37]). The initial evaluation of model fit was conducted using several indices along with their commonly accepted thresholds: the ratio of the Satorra–Bentler chi-square statistic to degrees of freedom (SB*χ*^2^/*df*) with values up to 5.00, Comparative Fit Index (CFI) and Tucker–Lewis Index (TLI), both at or above 0.95, Root Mean Square Error of Approximation (RMSEA) at or below 0.08, and Standardized Root Mean Square Residual (SRMR) not exceeding 0.08 (e.g., [45]; [50]). Additionally, internal consistency and factor quality were preliminarily assessed through Cronbach’s alpha (*α*) and McDonald’s omega (*ω*), with values of 0.70 or higher indicating acceptable reliability, the Latent Hancock’s H-index with a recommended minimum of 0.80 to evaluate latent variable replicability, and the Factor Determinacy Index (FDI), with values above 0.90 suggesting reliable factor score estimates.

All data analyses were performed utilizing SPSS version 29 ([46]) and RStudio software “https://www.r-project.org/ (accessed on 12 February 2025)”, employing the psych (v. 2.5.3), lavaan (v. 0.6.18), semTools (v. 0.5.6), and semPlots (v. 1.1.6) packages ([81]).

### 2.2. Results

#### 2.2.1. Phase 1: Initial Data Screening and Descriptive Overview

The means, standard deviations, skewness, kurtosis, item variance, item-to-total correlations, and item–item correlations are provided in detail in Table 1. The items of the Italian version of the MVTS showed skewness and kurtosis values within the acceptable range for a normal distribution (absolute values ranging from 0.19 to 1.00), with the exception of Item 1, which exhibited skewness and kurtosis values exceeding the recommended cut-off of 1.0 (−1.18 and 2.01, respectively). The Mardia’s test was not significant for skewness (*p* > 0.05) but was significant for kurtosis (*p* < 0.05), indicating a slight deviation from normality in the distribution. The adjusted item–total correlations ranged from 0.55 (item 1) to 0.79 (item 3), and the inter-item correlations were all statistically significant (*p* < 0.001) and above the threshold of 0.30 (min. 0.39, max. 0.72).

#### 2.2.2. Phase 2: Exploratory Factor Structure and Assessment of Preliminary Psychometric Properties

In relation to the initial assessment of factorability, The Kaiser–Meyer–Olkin (KMO) measure yielded a value of 0.898 (95% CI: 0.858, 0.909), indicating excellent sampling adequacy. Bartlett’s test of sphericity was statistically significant [*χ*^2^_(21)_ = 948.6, *p* < 0.001], and all individual Measures of Sampling Adequacy (MSA) spanned from 0.876 to 0.930. These findings support the suitability of the dataset for factor analysis and the appropriateness of the inter-item correlations for identifying latent constructs. Preliminarily, an Optimized Parallel Analysis supported a unidimensional factorial solution (see Table 2). EFA suggested a one-factor solution with the following model fit: CFI = 0.958, TLI = 0.936, RMSEA = 0.103 (90%; CI [0.074, 0.134]), SRMR = 0.039. Initial item inspection revealed that Item 1 (“I was exposed to distressing news and experiences via media”) exhibited inadequate factor loading (*λ* = 0.56) and communality (*h*^2^ = 0.34), both falling below recommended thresholds. Therefore, since its meaning appeared to be redundant with the content of item 2, this item was removed to achieve a more optimal solution, and the Exploratory Factor Analysis was re-conducted. The EFA confirmed a one-factor solution with an eigenvalue of 3.88, explaining 64.71% of the cumulative variance (see Table 2). This was further supported by a UniCo value of 0.988 (95% CI [0.958, 0.997]), an ECV value of 0.912 (95% CI [0.904, 0.945]), and a MIREAL value of 0.187 (95% CI [0.161, 0.205]). Following the removal of Item 1, a substantial improvement was also observed across all fit indices: CFI = 0.981, TLI = 0.968, RMSEA = 0.081 (90%; CI [0.041, 0.101]), SRMR = 0.027.

As reported in Table 2, all factor loadings were statistically significant, spanning a range from 0.66 (item 2) to 0.89 (item 5), and all items’ communalities ranged from 0.49 to 0.74. Finally, Cronbach’s alpha and McDonald’s omega were, 0.890 and 0.894, respectively, the H-Index was 0.911 [0.887, 0.925], and the FDI was 0.954.

## 3. Study II

### 3.1. Materials and Methods

#### 3.1.1. Sample Size Determination

To determine the optimal sample size, a power analysis was conducted using an a priori structural equation modeling calculator “https://www.danielsoper.com/statcalc/default.aspx (accessed on 10 June 2025)”, which recommended a minimum sample size of 200 to achieve a statistical power of 0.90 at a significance level of *p* < 0.01. As noted by [50] ([50]), the appropriate sample size for Confirmatory Factor Analysis (CFA) depends on the number of parameters that need to be estimated. In this study, CFA was conducted on a six-item scale, involving the estimation of 12 parameters—specifically six factor loadings and six error variances—associated with a standardized latent factor. Following [50]’s ([50]) guidelines, a minimum ratio of 10 participants per parameter is generally sufficient for stable CFA estimates, suggesting a required sample of 120. A stricter recommendation advises 20 participants per parameter, raising the threshold to 240. The final sample used in Study II exceeded both criteria, supporting the robustness and generalizability of the findings.

#### 3.1.2. Procedure and Participants

An initial pool of 565 respondents was screened for missing values and statistical outliers. Since no missing data were detected, the focus shifted to identifying multivariate outliers. Based on Mahalanobis distance values (*p* < 0.001), 12 cases were flagged as potential outliers, following established guidelines (e.g., [92]). These observations were subsequently reviewed and excluded to reduce the risk of bias in the main analyses.

The final sample consisted of 553 young adults, 340 (61.5%) females and 213 (38.5%) males, aged between 18 and 30 years (M = 22.43, SD = 2.37), predominantly from the Campania region (490, 88.6%). In terms of relationship status, 297 (53.7%) participants were single, while 256 (46.3%) were in a romantic relationship. Regarding educational attainment, 9 (1.6%) participants held a lower secondary school diploma, 416 (75.2%) held a higher secondary school diploma, 92 (16.6%) had a bachelor’s degree, 32 (5.8%) held a master’s degree, and 4 (0.7%) held a postgraduate qualification. Concerning occupational status, 322 (58.2%) participants were students, 134 (24.2%) were working students, 70 (12.7%) were employed, and 27 (4.9%) were unemployed. Additionally, 308 (55.7%) participants were enrolled in humanities-related university courses, while 149 (29.6%) were pursuing science-oriented courses. In terms of political orientation, 19 (3.4%) participants identified with right-wing ideologies, 46 (8.3%) identified with centrist parties, 262 (47.4%) identified with left-wing ideologies, and 226 (40.9%) participants declared no interest in any political orientation.

Recruitment for Study II was conducted between February and March 2025 through a snowball sampling strategy. As in the previous study, data were gathered via an online questionnaire shared across social media networks and promoted in commonly frequented areas within university settings. The eligibility criteria were consistent with those of Study I. Informed consent was obtained from all participants on the first page of the survey, prior to beginning the questionnaire.

#### 3.1.3. Measures

In addition to the Italian six-item version of the Media Vicarious Traumatization Scale (MVTS), the following instruments were administered:

A dedicated *socio-demographic section* was developed to gather detailed information about participants, including their age, gender, region of origin, marital status, educational background, employment status, area of university study, involvement in volunteer work, participation in charitable initiatives, religious affiliation, and political orientation.

The *War Worry Scale* (WWS, [70]) was used to assess worry about war through 10 items rated on a five-point Likert scale, ranging from 1 (not at all) to 5 (very much). This self-report instrument is designed to assess War Worry across two dimensions: Worry about the Present (WWP) and Worry about the Future (WWF). The WWP dimension captures immediate worries about ongoing conflicts, their intensity, and the impact on those directly involved. In contrast, the WWF dimension reflects apprehensions about the long-term consequences of war, including potential risks to the respondent’s future and loved ones, as well as fears of conflict escalation into broader or nuclear warfare. The instrument yields an overall score, with higher scores indicating greater levels of war-related worry. In the present study, the overall score was used and the WWS showed excellent internal consistency, with both Cronbach’s alpha of 0.91 and McDonald’s omega of 0.90.

An adapted version of the *War-related Media Exposure Scale* (WarMES, [29]) was used to measure the perceived frequency of exposure to distressing war-related informational content (news, images, videos) in recent weeks. The instrument consists of nine items rated on a five-point Likert scale (1 = Never, 5 = Daily). Example items include: “Destruction of civilian buildings (homes, hospitals, universities, etc.)”, “Civilians killed and victims trapped under rubble” and “Bombings and explosions”. As in the original version, the unidimensional scale yields a global score reflecting the perceived frequency of exposure to contemporary war-related information. The authors have demonstrated the good psychometric properties of the instrument, and in the present study, it showed a Cronbach’s alpha and McDonald’s omega of 0.92.

The *Generalized Anxiety Disorder Scale* (GAD-7, [90]) was used to assess generalized anxiety through seven items rated on a four-point Likert scale (ranging from 0 = not at all to 3 = nearly every day). This instrument has demonstrated strong criterion validity in identifying potential cases of Generalized Anxiety Disorder (GAD). The total score ranges from 0 to 21 and corresponds to different severity levels: 0–4 indicates minimal anxiety, 5–9 mild anxiety, 10–14 moderate anxiety, and 15–21 severe anxiety. Sample items include, for example, “Excessive worry about various things” and “Fear that something bad might happen.” In the current study, the GAD-7 showed excellent internal consistency, with both Cronbach’s alpha and McDonald’s omega equal to 0.89.

The *Dark Future Scale* ([47]) was used to evaluate future anxiety, with a focus on the emotional and cognitive mechanisms that foster fear and diminish a sense of hope. Responses are recorded on a seven-point Likert scale, ranging from 0 (completely false) to 6 (completely true). The total score can vary between 0 and 30, with higher values reflecting more intense anxiety regarding one’s future. Example of items include the following: “I worry that my current challenges will last a long time” and “I’m concerned that economic or political developments could negatively affect my future”. In this study, the scale exhibited excellent psychometric qualities, with both Cronbach’s alpha and McDonald’s omega reaching 0.91.

The *Life Orientation Test—Revised* (LOT-R; [85]; Italian adaptation by [4]) was employed to assess dispositional optimism using six items rated on a five-point Likert scale, ranging from 0 (strongly disagree) to 4 (strongly agree). Example items include statements such as “It’s hard for me to expect things will go well” and “If something might go wrong, it usually does in my life.” In the present study, the measure demonstrated acceptable internal consistency, with Cronbach’s alpha at 0.79 and McDonald’s omega at 0.77.

#### 3.1.4. Data Analysis

Preliminary descriptive statistics were computed to assess the means and standard deviations of each item on the MVTS. The distribution of the data was evaluated through the examination of skewness and kurtosis values, with values between −1 and +1 interpreted as indicative of approximate normality ([92]).

To verify the factorial structure, a Confirmatory Factor Analysis (CFA) was carried out. In line with the results from Study I, a unidimensional factorial model was estimated using the robust maximum likelihood method (MLM), which yields robust standard errors and employs the Satorra–Bentler chi-square statistic (SB*χ*^2^) for model evaluation.

Model fit was evaluated using the same criteria as in Study I. Specifically, the following indices and their recommended thresholds were considered: a ratio of SBχ^2^ to degrees of freedom (SB*χ*^2^/*df*) of 5.00 or lower; a Comparative Fit Index (CFI) of at least 0.90 for an acceptable fit and 0.95 for a good fit; a Tucker–Lewis Index (TLI) of at least 0.90 for an acceptable fit and 0.95 for a good fit; a Root Mean Square Error of Approximation (RMSEA) of less than 0.05 for a good fit and less than or equal to 0.08 for an acceptable fit; and a Standardized Root Mean Square Residual (SRMR) of less than 0.05 for a good fit and less than or equal to 0.08 for an acceptable fit ([45]; [50]).

Measurement invariance (MI) analyses were conducted to assess whether the factorial structure remained stable across gender (males vs. females). Initially, model fit was examined separately within each subgroup using the fit indices and cutoffs previously described. Subsequently, a series of increasingly constrained nested models were estimated to evaluate invariance across gender. The first model tested configural invariance, in which the factor structure was freely estimated in both groups (Model 1). The second model tested metric invariance, with factor loadings constrained to be equal across groups (Model 2). The third model tested scalar invariance, constraining both factor loadings and item intercepts to equality (Model 3). In cases where full scalar invariance was not supported, partial scalar invariance was tested by relaxing the equality constraints on intercepts for specific items, following established guidelines suggested by [17] ([17]). Fit for each model was assessed using the same criteria: SB*χ*^2^/*df* ≤ 5.00, CFI and TLI ≥ 0.90 for good fit and ≥ 0.95 for excellent fit, and RMSEA and SRMR ≤ 0.08 for acceptable fit and ≤ 0.05 for good fit. To evaluate invariance, differences in fit between nested models were examined using the Satorra–Bentler scaled chi-square difference test (DIFFTEST), changes in CFI (|ΔCFI|), and changes in RMSEA (|ΔRMSEA|). Given the sensitivity of the chi-square statistic to sample size, |ΔCFI| (≤0.01) and |ΔRMSEA| (≤0.015) were used as the primary criteria ([22]). According to [21] ([21]), a meaningful decline in model fit is indicated when at least two of the three thresholds are exceeded.

The reliability of the MVTS was assessed using both Cronbach’s alpha (*α*) and McDonald’s omega (*ω*) coefficients. To preliminarily evaluate convergent validity, Composite Reliability (CR; values ≥ 0.70), Average Variance Extracted (AVE; values ≥ 0.50), and item factor loadings (*λ* ≥ 0.50) were examined, based on the criteria proposed by [32] ([32]). Further evidence for convergent validity was gathered by calculating Pearson correlation coefficients between the MVTS and theoretically related psychological constructs, including worry about war, exposure to distressing war-related content through the media, generalized anxiety, and future anxiety.

Discriminant validity was assessed by comparing the square root of the AVE (SQRT-AVE) for the MVTS with its correlation coefficients with worry about war, exposure to distressing war-related media content, and intolerance of uncertainty. According to [32]’s ([32]) guideline, discriminant validity is supported when the square root of the AVE (SQRT-AVE) for each construct exceeds the correlation between the constructs, or when no significant association is observed.

To supplement traditional reliability estimates, analyses based on Item Response Theory (IRT) were performed using the Graded Response Model (GRM; [82]), which is suitable for items with ordered categorical responses. For each item, discrimination parameters (a) and thresholds (*b*_1_–*b*_4_) were estimated to evaluate the item’s capacity to differentiate among varying levels of the latent trait and to identify the points along the trait continuum where transitions between response categories occur. Discrimination values greater than 1.70 were considered indicative of good item performance. Item Information Curves (IICs) were used to explore how much information each item provides across different levels of *θ*, while the Test Information Function (TIF) offered insight into the overall precision of the scale. A Conditional Reliability Curve was generated to assess measurement precision as a function of *θ*. Both the median and maximum reliability values were reported, and interpretability was guided by a conventional cut-off of 0.80, indicating acceptable precision. In addition, the marginal reliability of the scale was computed to provide an overall estimate of reliability across the trait distribution. These analyses offered a more refined perspective on the psychometric properties of the instrument, complementing traditional methods with item- and trait-level precision metrics ([7]).

Finally, prior to conducting the analyses of predictive and incremental validity, a preliminary analysis was carried out using an independent-samples *t*-test to examine whether high and low levels of Media Vicarious Traumatization (MVTS) were associated with significant differences in levels of generalized anxiety and future anxiety. For this purpose, the sample was split into two groups based on the median MVTS score, distinguishing between low and high levels. Then, predictive and incremental validity were examined using two hierarchical regression analyses, following preliminary assessments of tolerance and residual diagnostics. In each model, generalized anxiety (Model 1) and future anxiety (Model 2) served as the dependent variables. Worry about war (WWS) was entered in Step 1, followed by Media Vicarious Traumatization (MVTS) in Step 2, allowing the unique predictive contribution of MVTS to be assessed beyond the variance explained by worry about war.

Statistical Analyses were performed utilizing SPSS version 29 ([46]) and RStudio software “https://www.r-project.org/ (accessed on 10 April 2025)”, employing the lavaan (v. 0.6.18), semtools (v. 0.5.6), mirt (v. 1.44.0), psych (v. 2.5.3), semplots (v. 1.1.6), and ggplot2 (v.3.5.1) packages ([81]).

### 3.2. Results

#### 3.2.1. Phase 1: Confirmatory Factor Analysis

All descriptive statistics are reported in Table 3. The findings indicated that both individual items and the overall scale followed a normal distribution. In line with EFA results, the unidimensional factor model showed good fit of the data: SB*χ*^2^(9) = 40.529; *p* < 0.01; SB*χ*^2^/*df* = 4.503; CFI = 0.973; TLI = 0.953; RMSEA = 0.080 (90% CI [0.060–0.094]); SRMR = 0.031. As shown in Table 2 and illustrated in Figure 1, all factor loadings reached statistical significance, with values ranging from 0.646 (item 2) to 0.864 (item 5).

#### 3.2.2. Phase 2: Assessment of Measurement Invariance Across Gender

In examining measurement invariance of the MVTS across gender groups (male vs. female), the scale exhibited a good model fit within the male sample: SBχ^2^(9) = 30.263, *p* < 0.001; SB*χ*^2^/*df* = 3.362; CFI = 0.961; TLI = 0.934; RMSEA = 0.083, 90% CI [0.069–0.093]; SRMR = 0.041. Likewise, the female sample showed similarly adequate model fit indices: SB*χ*^2^(9) = 24.915, *p* < 0.001; SBχ^2^/*df* = 2.768; CFI = 0.969; TLI = 0.948; RMSEA = 0.072, 90% CI [0.046–0.091]; SRMR = 0.037.

In the preliminary analysis, the configural invariance model applied to the entire sample revealed a robust fit: SB*χ*^2^(18) = 54.404, *p* < 0.001; SB*χ*^2^/*df* = 3.022; CFI = 0.966; TLI = 0.943; RMSEA = 0.086, 90% CI [0.064–0.097]; SRMR = 0.034. These results suggest that the factor structure was equivalent between males and females.

In the second phase, the metric invariance model was applied and demonstrated a good fit: SB*χ*^2^(23) = 63.232, *p* < 0.001; SB*χ*^2^/*df* = 2.749; CFI = 0.962; TLI = 0.950; RMSEA = 0.080, 90% CI [0.060–0.0928]; SRMR = 0.046. The results showed a non-significant reduction in the DIFFTEST (ΔSB*χ*^2^ = 7.034; *df* = 5; *p* = 0.218), along with a non-significant decrease in |ΔCFI| = 0.004 and |ΔRMSEA| = 0.006, suggesting that the items were equally associated with the latent factor, regardless of gender.

In a subsequent step, the scalar model was tested. However, unlike the metric model, full scalar invariance could not be established, suggesting that at least some responses differ significantly between the groups. Consequently, a partial scalar model was specified. To identify which items to free, the lavTestScore function was used, which provided insights into the parameters showing significant differences across groups. Based on this analysis, the intercepts for Item 2 and Item 6 were identified as non-invariant and were freed in the partial scalar model, while all other parameters remained invariant. This partial scalar model demonstrated good fit: SB*χ*^2^(26) = 69.366, *p* < 0.001; SB*χ*^2^/*df* = 2.668; CFI = 0.959; TLI = 0.953; RMSEA = 0.078, 90% CI [0.059–0.087]; SRMR = 0.048. The comparison between the partial scalar model and the metric model showed no significant differences in the DIFFTEST (ΔSB*χ*^2^ = 5.495; *df* = 3; *p* = 0.139), with ΔCFI = 0.003 and ΔRMSEA = 0.002, suggesting that partial invariance was deemed acceptable, thus allowing for a valid comparison between the groups. Table 4 summarizes the results of the model comparisons for gender invariance.

#### 3.2.3. Phase 3: Assessment of Internal Consistency and Convergent and Divergent Validity

Regarding internal consistency, Cronbach’s alpha and McDonald’s omega were 0.884 and 0.886, respectively, reflecting a high level of reliability. Additionally, each item showed a significant contribution to the overall internal coherence of the MVTS.

An initial assessment of convergent validity revealed that all factor loadings exceeded the 0.50 threshold (refer to Table 3 and Figure 1), with a Composite Reliability (CR) of 0.885 and an Average Variance Extracted (AVE) of 0.748. Moreover, the Pearson correlations presented in Table 4 showed positive and significant associations between MVTS and Generalized Anxiety (GAD), Future Anxiety (DFS), Worry about War (WWS), and exposure to distressing war-related content (WarMES), indicating adequate convergent validity of the instrument.

Discriminant validity was primarily assessed by comparing the square root of the AVE of the MVTS (SQRT-AVE = 0.865) with its correlations with the WWS, WarMES, and IUS scales. As shown in Table 5, the correlations between the MVTS and both WWS and ESP-W were lower than the SQRT-AVE of the MVTS, as expected according to the [32] ([32]) criterion. These results supported the notion that the construct of Media Vicarious Traumatization—examined in this study in relation to war-related information—was distinct from mere exposure to distressing war content, worry about war, and intolerance of uncertainty, despite the presence of positive and significant associations. Additionally, the non-significant negative correlation between MVTS and LOT-R further supported the discriminant validity of the scale, indicating that the construct was not associated with dispositional traits such as optimism or pessimism.

#### 3.2.4. Phase 4: Item Response Theory (IRT) Analyses of the Media Vicarious Traumatization Scale (MVTS)

As reported in Table 6, the IRT analysis produced discrimination parameters (*a*) ranging from 1.793 to 3.992 across the six items, demonstrating strong sensitivity of all items to variations in the latent trait. All items exhibited good to excellent discrimination, thereby confirming the overall effectiveness of the scale’s items. Threshold parameters (*b*_1_ to *b*_4_) were distributed over a wide range, spanning from −3.297 to 1.198. This suggests that the items cover a broad spectrum of difficulty levels, allowing the scale to effectively assess individuals from low to high levels of the latent construct. For instance, Item 2 exhibited the lowest threshold (*b*_1_ = −3.297), indicating sensitivity of respondents with very low trait levels, while Item 3 showed the highest threshold (*b*_4_ = 1.198), representing items targeted at higher trait levels.

As reported in Figure 2, analysis of the Item Information Curves (IIC) revealed that the scale provides a reliable measurement of vicarious traumatization across a broad range of the latent trait (*θ*). Most items (Items 1–4) offer peak information around the average range of *θ*, where individuals are most commonly situated, supporting the scale’s sensitivity to typical levels of vicarious trauma. Item 5 contributes valuable information at lower trait levels, while Item 6 shows a broader curve, ensuring coverage across a wider trait spectrum. Overall, the distribution of information suggests that the items function well collectively, enhancing the precision of measurement and supporting the scale’s psychometric adequacy.

As reported in Figure 3, the Test Information Function (TIF) demonstrated that the MVTS provides high information within a moderate range of the latent trait, specifically between approximately *θ* = −2.5 and *θ* = 1. This pattern aligns with broader trends in the development of psychological scales, particularly those designed for non-clinical populations, as in the present study. Given that most respondents in such samples tend to fall within the mid-range of psychological constructs, the scale appears appropriately calibrated to maximize precision where it is most needed. The lower precision observed at the extremes of the trait reflects a deliberate calibration strategy, aimed at capturing meaningful variation in typical levels of vicarious traumatization rather than focusing on extreme cases. This indicates that the scale is particularly effective in capturing variations in media vicarious traumatization, especially at lower to moderate trait levels, supporting its usefulness in diverse non-clinical populations.

Regarding IRT-based reliability, as reported in Figure 4, findings showed generally satisfactory values across the latent trait continuum, with a median reliability reaching 0.87 and a maximum reliability of 0.91. Reliability stays above the common threshold of 0.80 throughout the central trait range, confirming the test’s suitability for accurately assessing individuals within this range. Additionally, the marginal reliability of 0.893 indicates strong overall consistency of the test scores across the entire sample.

#### 3.2.5. Phase 5: Assessment of Predictive Validity and Incremental Utility

Preliminary, independent-samples *t*-tests conducted using a median split of the Media Vicarious Traumatization Scale (MVTS; Median = 3.83) revealed significant differences in both generalized anxiety (GAD) and future anxiety (DFS) between participants with high versus low MVTS scores. Specifically, individuals with higher MVTS levels reported significantly greater generalized anxiety (M*_HIGH_* = 12.45 vs. M*_LOW_* = 9.69; *t*(551) = 6.30, *p* < 0.001, Cohen’s *d* = 0.54) and future anxiety (M*_HIGH_* = 4.34 vs. M*_LOW_* = 3.35; *t*(551) = 8.52, *p* < 0.001, Cohen’s *d* = 0.73) compared to those with lower MVTS scores.

Regarding preliminary assumptions for the hierarchical regression analyses, tolerance values ranged from 0.612 to 0.627, and Variance Inflation Factor (VIF) values ranged from 1.000 to 1.604, indicating no issues with multicollinearity among the predictors. Additionally, Durbin–Watson statistics for the two regression models ranged from 1.842 to 2.027, suggesting the absence of significant autocorrelation in the residuals.

Table 7 presents the results concerning the predictive validity and incremental value of the MVTS in relation to two outcome variables: Generalized Anxiety (Model 1) and Future Anxiety (Model 2). All models were statistically significant (*p* < 0.001), with the MVTS consistently emerging as a strong and significant predictor across both outcomes. In Model 1, WWS was entered in the first step, explaining 10.4% of the variance in Generalized Anxiety. The addition of the MVTS in the second step led to a statistically significant increase in explained variance, raising the total to 12.2%. This reflects an approximate increase of 20.2% over the baseline model, supporting the incremental predictive value of the MVT beyond the initial predictor. Similarly, in Model 2, WWS accounted for 17.7% of the variance in Future Anxiety in the first step. Including the MVTS in the second step increased the explained variance to 23.2%, corresponding to an approximate gain of 31.6%. These findings further underscore the unique contribution of the MVTS in predicting Future Anxiety above and beyond WWS. Finally, based on standardized coefficients (*β*) of Model 2, the MVTS proved to be a stronger predictor than the WWS.

## 4. Discussion

This study describes the process of cultural adaptation and psychometric validation of the Media Vicarious Traumatization Scale (MVTS), a six-item scale designed to measure the emotional and cognitive effects that may develop following persistent and cumulative exposure to traumatic media content. The adaptation and validation process of the MVTS followed the phases recommended in the literature ([14]) and consisted of two distinct but interconnected studies that led to the final version of the instrument, which is provided in Appendix A, along with its scoring procedure.

Study I describes the back-translation process that, starting from the English-adapted version by [54] ([54]), led to the Italian version of the instrument. Subsequently, Study I explored the descriptive statistics of the individual items, which not only guided the subsequent analyses related to the exploration of the latent structure but also highlighted that all items played a significant role in defining the construct, with item-to-item and item-to-total correlations exceeding 0.30, in accordance with Classical Test Theory ([14]). Parallel Analysis (PA) and Exploratory Factor Analysis (EFA) suggested the unidimensional latent structure of the MVTS, and item inspection led to the removal of Item 1 due to factor loading and communality values below the recommended cut-offs ([37]). The resulting six-item scale, re-examined through EFA, confirmed a unidimensional structure explaining 64.71% of the variance of the extracted factor and showed that all factor loadings and item communalities exceeded the recommended thresholds ([37]). The instrument demonstrated good preliminary psychometric properties, with excellent internal consistency and high scores on the Latent Hancock’s Index (H-Index) and Factor Determinacy Index (FDI), indicating good factor accuracy and stability.

Study II, conducted on an independent sample of Italian young adults, corroborated the results from the EFA and PA of Study I. Confirmatory Factor Analysis (CFA) supported the unidimensional structure of the instrument, yielding good fit indices consistent with established scientific guidelines ([45]; [50]). The CFA showed that all items had factor loadings above the 0.60 cutoff, playing a significant role in defining the construct and explaining its variance (see Table 2).

The instrument demonstrated measurement invariance across gender, showing good model fit for both the female and male subgroups. In line with [22]’s ([22]) recommendations, and as schematically reported in Table 3, the MVTS proved capable of maintaining the same factorial structure across the two tested subgroups (configural invariance). Furthermore, the absence of significant differences in factor loadings between the two subgroups suggests metric invariance, indicating no gender differences in the attribution of meaning to the construct. Full scalar invariance was not achieved, indicating that the two subgroups responded differently to some items. Accordingly, in agreement with [17] ([17]), significant violations were identified in the intercept constraints of Items 2 and 6, which were subsequently freed. The partial invariance model showed good fit and, as reported in Table 3, was not significantly worse than the metric invariance model, allowing for mean comparisons between genders. These results align with the existing literature highlighting a greater tendency toward internalization among women ([79]), which, as suggested by our findings, may lead women to respond differently to certain items compared to men, thereby generating variation in the expected mean values of the identified items.

The instrument demonstrated excellent internal consistency, surpassing that of the original seven-item version by [54] ([54]) and aligning with the more recent version used by [53] ([53]), thereby confirming its reliability. In accordance with the criteria outlined by [32] ([32]), the scale exhibited good internal and convergent validity. All items showed factor loadings above 0.50, and the instrument achieved a high Composite Reliability score, further supporting the internal consistency findings. The Average Variance Extracted (AVE) index of 0.748, exceeding the 0.50 cutoff, indicates that a substantial proportion of the total variance in the observed items is explained by the latent construct, preliminarily confirming the good convergent validity of the MVTS. Significant positive correlations (see Table 4) between the MVTS and the Generalized Anxiety Disorder (GAD) scale, the Dark Future Scale (DFS), the War Worry Scale (WWS), and the War-related Media Exposure Scale (WarMES) further support the convergent validity of the instrument. The positive associations between MVTS, anxiety (GAD), and anxious future outlook (DFS) align with the existing literature highlighting the link between media vicarious traumatization and various forms of psychological distress in both general and clinical populations ([53]; [54]; [96]; [97]). These findings confirm the instrument’s concurrent validity through its relationship with mental health outcomes. Moreover, since in our study the MVTS was specifically calibrated to contemporary war-related information, the observed positive associations with the War Worry Scale (WWS) and the War-related Media Exposure Scale (WarMES) were hypothesized a priori and confirm the convergent validity of the MVTS. These results are consistent with studies demonstrating the impact of exposure to ongoing war-related media content on mental health ([55]; [30]).

Despite the positive associations with the War Worry Scale (WWS), the War-related Media Exposure Scale (WarMES), and the Intolerance of Uncertainty Scale (IUS), the comparison between the square root of the Average Variance Extracted (AVE) of the MVTS and its correlation coefficients with these measures confirmed its discriminant validity (see Table 4). These results indicate that—despite significant positive correlations—Media Vicarious Traumatization is a distinct construct, not overlapping with Worry about War ([70]), the dispositional factor of intolerance of uncertainty (IUS), or the frequency of exposure to anxiety-inducing content, as measured by the WarMES. Finally, the lack of a significant correlation between the MVTS and the Life Orientation Test—Revised (LOT-R), which assesses the dispositional trait of optimism ([85]), further supports the discriminant validity of the MVTS. This aligns with the original conceptualization of vicarious trauma ([66]), indicating that the MVTS measures situational psychological distress related specifically to media exposure.

The IRT analysis provides solid evidence for the psychometric quality of the adapted scale. All items demonstrated good discriminatory capacity across different levels of Media Vicarious Traumatization, and the wide distribution of threshold parameters indicates that the scale is capable of capturing a broad spectrum of the construct. Item-level information curves showed that most items are particularly informative around average levels of the trait—where non-clinical populations typically fall—while others extend the coverage to lower or broader ranges of the continuum. The test information function confirms high reliability within the moderate range of the trait, aligning with the scale’s aim to assess common (i.e., non-clinical) levels of vicarious traumatization in youth populations rather than extreme manifestations. These findings highlight the adequacy of the instrument and are consistent with those emerging from classical reliability analyses, further reinforcing its psychometric robustness.

In order to enrich the correlational analyses, the Media Vicarious Traumatization Scale (MVTS) scores were split at the median and compared through an independent-samples *t*-test. This allowed the identification of significant differences between individuals with high versus low media vicarious traumatization in terms of generalized anxiety and anxious future outlook. This finding guided the development of two hierarchical regression models, which, as reported in Table 5, confirm the predictive and incremental validity of the MVTS. These models were constructed to explore whether and to what extent war-related worry and media vicarious traumatization—fueled by news, images, and videos about the war—contribute to increased generalized anxiety and catastrophic future expectations among Italian young adults. In both models, war-related worry and media vicarious traumatization emerged as significant predictors of higher anxiety levels and anxious future outlook. While the impact of war-related worry on young adults’ mental health has already been investigated ([72], [70]), the results concerning the role of war-calibrated media vicarious traumatization are entirely novel, albeit consistent with previous studies highlighting the construct’s impact on psychological well-being ([54]; [96]; [97]). The regression models confirm that media vicarious traumatization is a significantly relevant variable in assessing the impact of contemporary wars on young people’s mental health, substantially increasing the explained variance of the two outcomes. Our findings align with recent research documenting the effects of media exposure to war on various forms of mental distress, including depressive and anxiety symptoms, sleep disturbances, and emotional distress ([98]; [55]; [30]; [60]). Consistent with the recent contribution by [53] ([53]), our results underscore the importance of exposure to highly traumatic media events for psychological well-being, particularly among young adults who, due to their continuous and often pervasive use of social media platforms driven by recommendation algorithms ([20]; [61]), represent a population at risk for media overexposure and, consequently, media vicarious traumatization.

### Limitations and Future Directions

This study presents several limitations that warrant consideration. First, the use of a convenience sampling method and reliance on self-report measures may have introduced biases related to participants’ individual characteristics or the tendency to respond in socially desirable ways, potentially affecting the validity of the findings. Additionally, the adaptation and validation process of MVTS was conducted within a relatively homogenous sample, primarily composed of young adults with high educational attainment. Broader validation efforts are needed, involving more diverse and representative populations across varying age groups, educational backgrounds, and employment statuses. Moreover, participants were predominantly recruited from a single Italian region (Campania), and the sample showed an overrepresentation of students from the humanities. Future studies should therefore aim to replicate the research with more diverse populations. Such future research would facilitate a more comprehensive evaluation of the Italian version of the MVTS and its psychometric properties. Moreover, the study’s cross-sectional design restricts the ability to assess changes in Media Vicarious Traumatization over time. Longitudinal studies are recommended to examine the temporal stability and predictive utility of the Italian MVTS across different life stages. Future studies could further investigate the predisposing factors of Media Vicarious Traumatization, focusing, for instance, on dispositional personality traits. Additionally, they could explore the potential mediating and/or moderating role that this construct may play in order to gain a deeper understanding of its impact on mental health.

## 5. Conclusions and Practical Implications

This study provides a valid, robust, and reliable instrument for assessing Media Vicarious Traumatization (MVT), a construct highlighted in the literature as a potential effect of exposure to distressing content disseminated via mass media and social media. Specifically, in this work, the scale is calibrated to the context of war through a tailored instruction set. It is important to emphasize that the instrument’s versatility allows it to be adapted, by modifying the instructions, to other potentially traumatic events such as extreme climate phenomena, which have also been identified as factors potentially impacting psychological well-being ([18]; [73], [76]).

The present work consists of two studies, which, filling a gap in the literature on the topic, present the adaptation and validation process of the Media Vicarious Traumatization Scale (MVTS). The study results, stemming from the integration of Classical Test Theory (CTT) and Item Response Theory (IRT) for the exploration of psychometric properties, confirm the factorial robustness, reliability, measurement invariance across gender (configural, metric, and partial scalar), and, finally, the convergent, discriminant, and predictive validity of the instrument.

The MVTS represents a novel measure to detect the psychological impact of media exposure and, in an era marked by collective traumatic events such as ongoing wars, addresses the need to investigate the role that mass media and social media play in fostering forms of mental suffering. Indeed, the results of Study II clearly demonstrate that war-calibrated MVT is a central construct for exploring the psychological impact of conflicts on populations not directly involved. It plays a significant predictive role in generalized anxiety and anxious future outlook among the recruited young adults. This finding, highlighting the influence that contemporary collective dimensions have on youth distress, offers, in our view, a deeper understanding and clearer delineation of the contours of contemporary youth malaise, which is widely documented in the literature ([63]; [102]; [86]; [75]).

We believe this new instrument is particularly valuable in both individual and group clinical intervention settings, as it enables the assessment of whether—and to what extent—exposure to distressing media content contributes to psychological distress. While this aspect provides valuable insight for clinical practice, especially with youth populations, the MVTS can also play a central role in guiding the development of group-based psychological support interventions in non-clinical contexts. Within group support settings, the MVTS can help increase awareness and critical thinking about the use of mass media and social media, especially among young adults and adolescents who are heavily and pervasively exposed to them. Given the instrument’s versatility, it could be adapted as needed to potentially traumatic contemporary events that are algorithmically promoted as trending topics on social networks (e.g., wars, climate disasters, femicides). We maintain that supporting a healthier and more functional use of media and social media—aimed at selecting information sources based on their ability to help individuals understand the events to which they are exposed—could reduce feelings of helplessness and the negative psychological effects of pathological media overexposure, thereby mitigating its impact on psychological well-being. Finally, the MVTS could prove useful not only in the implementation of psychoeducational interventions but also for the longitudinal evaluation of their effectiveness.

## Figures and Tables

**Figure 1 ejihpe-15-00184-f001:**
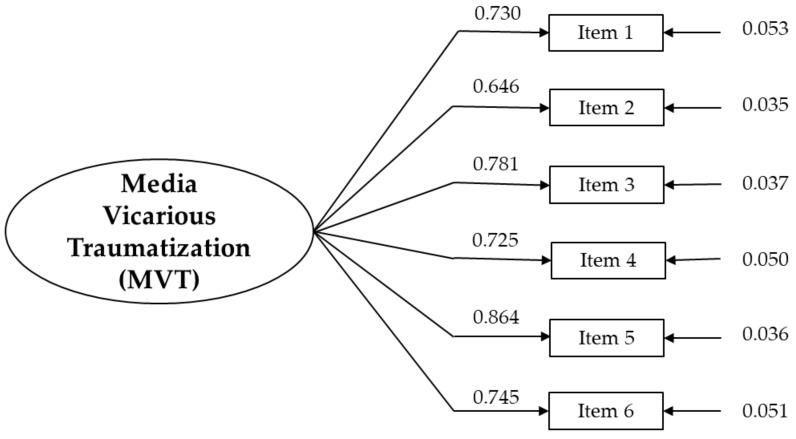
Graphical representation of the MVTS.

**Figure 2 ejihpe-15-00184-f002:**
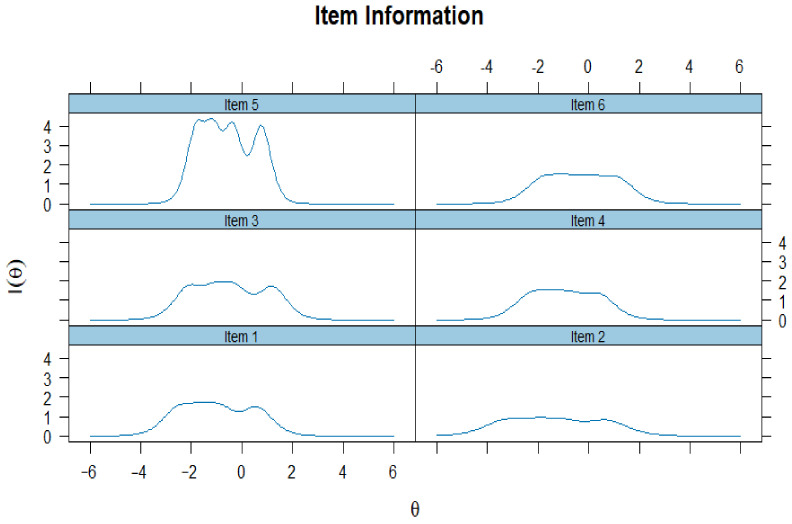
Graphical representation of IIC of MVTS.

**Figure 3 ejihpe-15-00184-f003:**
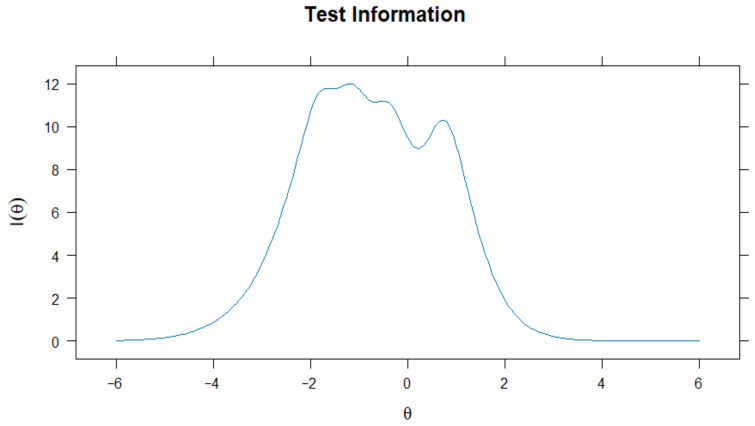
Graphical representation of TIF of MVT.

**Figure 4 ejihpe-15-00184-f004:**
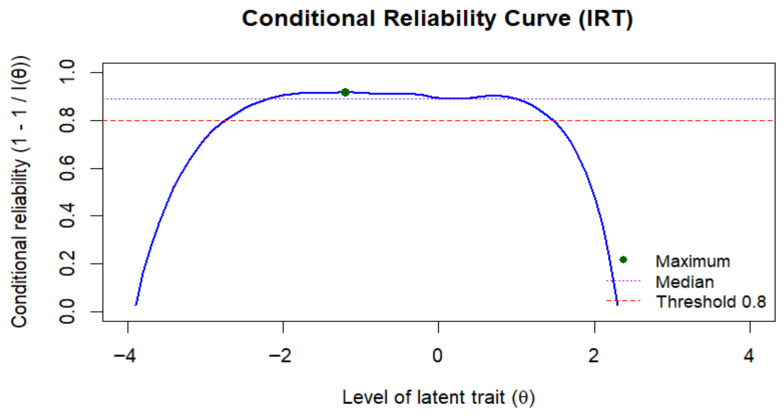
Graphical representation of IRT Conditional Reliability Curve.

**Table 1 ejihpe-15-00184-t001:** Descriptive statistics for items of MVTS (N = 250).

Item	M	SD	Var.	Sk	K	*r*-tot.	1	2	3	4	5	6	7
**1**	3.97	0.77	0.78	−1.18	2.01	0.55	-						
**2**	3.88	1.02	1.04	−0.91	0.28	0.74	0.58	-					
**3**	4.00	0.86	0.74	−0.98	1.00	0.63	0.42	0.53	-				
**4**	3.44	1.01	1.02	−0.44	−0.52	0.69	0.39	0.58	0.46	-			
**5**	3.89	1.07	1.14	−0.86	0.19	0.71	0.44	0.56	0.55	0.59	-		
**6**	3.60	1.09	1.20	−0.55	−0.31	0.79	0.47	0.63	0.51	0.64	0.66	-	
**7**	3.28	1.14	1.29	−0.20	−0.72	0.71	0.39	0.58	0.51	0.56	0.54	0.72	-

**Note**: M = Mean; SD = Standard Deviation; Var. = Variance; Sk = Skewness; K = Kurtosis; *r*-tot. = Adjusted Item-to-Total Correlation; all item correlations were significant (*p* < 0.001).

**Table 2 ejihpe-15-00184-t002:** Optimized Parallel Analysis (PA) based on Minimum Rank Factor Analysis, Explorative Factor Analysis (EFA), and Item Inspection.

Optimized Parallel Analysis (PA)				Item Inspection		
Factor	Real-Data Eigenv.	Mean of Random Eigenvalues	95% of Random Eigen.	Item	*λ*	*h* ^2^
				1	0.76	0.58
1	71.06 *	28.90	32.29	2	0.66	0.49
				3	0.74	0.55
**Exploratory Factor Analysis (EFA)**				4	0.76	0.58
**Factor**	**Eigenvalue**	**Prop. of Variance**	**Cum. Prop. of Variance**	5	0.86	0.74
1	4.37 *	64.71	64.71	6	0.77	0.59

**Notes:** PA = * Advised number of dimensions: 1; Number of random correlation matrices: 500; Method for random correlation matrices: Permutation of raw data. EFA = * Eigenvalue > 1.0; λ = standardized factor loadings; *h*^2^ = communalities.

**Table 3 ejihpe-15-00184-t003:** Item Descriptive Statistics and Confirmatory Factor Analysis (N = 553).

	Descriptive Analysis				CFA		
Item	M	SD	Sk	K	*λ*	95% CI	*R* ^2^
**1**	3.94	0.98	−0.93	0.53	0.730	0.633–0.788	0.533
**2**	4.00	0.88	−0.87	0.72	0.646	0.520–0.659	0.417
**3**	3.52	1.06	−0.55	−0.37	0.781	0.758–0.892	0.610
**4**	3.97	1.06	−0.99	0.45	0.725	0.668–0.857	0.525
**5**	3.67	1.09	−0.68	−0.11	0.864	0.859–0.964	0.747
**6**	3.32	1.17	−0.27	−0.71	0.745	0.786–0.943	0.556
**MVTS**	3.74	0.83	−0.60	−0.60	-	-	-

**Note:** M = mean; SD = standard deviation; Sk = skewness; K = kurtosis; CFA = confirmatory factor analysis; MVTS = global score. In CFA columns, absolute values of standardized factor loading (|*λ*|) are reported. *λ* = factor loading onto the specific factor, all *λ* values are statistically significant with *p* < 0.001; 95% CI = Confidence Intervals for standardized factor loadings; *R*^2^ = variance explained.

**Table 4 ejihpe-15-00184-t004:** Models Comparison for Measurement Invariance across Gender.

Models	SB*χ*^2^ (*df*)	CFI	RMSEA	Comparison	DIFF*χ*^2^ (*df*)	|ΔCFI|	|ΔRMSEA|
**Configural Inv.**	54.404 * (18)	0.966	0.086	-	-	-	-
**Metrical Inv.**	63.232 * (23)	0.962	0.080	Conf. vs. Met.	7.034 (5)	0.004	0.006
**Partial Scalar Inv.**	69.366 * (26)	0.959	0.078	Met. vs. Scal.	5.495 (3)	0.003	0.002

**Note**: * = *p* < 0.001; SB*χ*^2^ = Satorra–Bentler scaled chi-squared test; *df* = degrees of freedom; Δ = differences between indices; CFI = comparative fit index; RMSEA = root mean square error of approximation.

**Table 5 ejihpe-15-00184-t005:** Pearson Correlations for Convergent Validity and Divergent Validity of MVTS.

	AVE	1	2	3	4	5	6	7
**1. MVTS**	0.748	-						
**2. WWS**	0.788	0.614 *	-					
**3. WARMES**	0.770	0.427 *	0.384 *	-				
**4. DFS**	0.812	0.444 *	0.421 *	0.218 *	-			
**5. IUS**	0.501	0.274 *	0.325 *	0.131 *	0.540 *	-		
**6. GAD**	0.737	0.312 *	0.323 *	0.180 *	0.500 *	0.478 *	-	
**7. LOT-R**	0.603	−0.053	−0.045	−0.008	−0.078	−0.048	−0.073	-

**Note**: AVE = Average Variance Extracted; MVTS = Media Vicarious Traumatization Scale; WWS = War Worry Scale; WarMES = War-related Media Exposure Scale; DFS = Future Anxiety; IUS = Intolerance of Uncertainty; GAD = Generalized Anxiety; LOT-R = Life Orientation Test—Revised; * *p* < 0.001.

**Table 6 ejihpe-15-00184-t006:** Item Parameter Estimates from the Graded Response Model (IRT) for the MVTS.

Item	*a*	*b* _1_	*b* _2_	*b* _3_	*b* _4_
1	2.422	−2.499	−2.499	−0.864	0.585
2	1.793	−3.297	−3.297	−1.040	0.715
3	2.603	−2.096	−2.096	−0.280	1.198
4	2.272	−2.283	−2.283	−0.740	0.403
5	3.992	−1.798	−1.798	−0.362	0.768
6	2.292	−0.1.792	−1.792	0.089	1.145

**Note:** *a* = item discrimination parameter; *b*_1_*–b*_4_ = threshold (location) parameters indicating the points on the latent trait (*θ*) where the probability of endorsing the next response category becomes greater than the previous one.

**Table 7 ejihpe-15-00184-t007:** Hierarchical Regression Analyses for Predictive and Incremental Utility of MVTS.

Models/Step	Predictors	Adj. *R*^2^	Δ*R*^2^	ΔF	*β* Model	*t*
Model 1: Generalized Anxiety						
Step 1	WWS	0.104 *	0.104 *	64.160	0.211 *	4.185
Step 2	MVTS	0.122 *	0.021 *	12.948	0.182 *	3.598
Model 2: Future Anxiety						
Step 1	WWS	0.177 *	0.177 *	118.442	0.237 *	2.254
Step 2	MVTS	0.232 *	0.056 *	38.811	0.299 *	4.501

**Note**: Step 1 = WWS; Step 2 = WWS + MVTS; WWS = War Worry Scale; MVTS = Media Vicarious Traumatization Scale. All beta effects were standardized; * *p* < 0.001.

## Data Availability

The data supporting the findings of this study are available from the corresponding author upon reasonable request.

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
