# Peer review of "Media Exposure and Vicarious Trauma: Italian Adaptation and Validation of the Media Vicarious Traumatization Scale and Its Impact on Young Adults’ Mental Health in Relation to Contemporary Armed Conflicts"

_ejihpe, 2025, doi:10.3390/ejihpe15090184_

Round 1

Reviewer 1 Report

Comments and Suggestions for Authors

Thank you for the opportunity to revise the manuscript entitled “Media Exposure and Vicarious Trauma: Italian Adaptation and Validation of the Media Vicarious Traumatization Scale and Its Impact on Young Adults’ Mental Health in Relation to Contemporary Armed Conflicts”.

The manuscript presents the Italian adaptation and psychometric validation of the Media Vicarious Traumatization Scale (MVTS), calibrated on the context of contemporary armed conflicts. Across two studies, the authors examine the scale's structure, reliability, validity and measurement invariance across gender. The study includes a total sample of 803 Italian young adults (aged 18–30) and provides evidence for the MVTS’s psychometric soundness.

I believe the study covers a very relevant and timely topic by operationalizing the concept of media vicarious traumatization. The use of both classical test theory and item response theory (IRT) significantly enhances the robustness of the findings. The scale demonstrates strong reliability and validity metrics, and the application of measurement invariance analysis confirms its stability across gender groups. Overall, the MVTS emerges as a useful tool for detecting psychological vulnerability in young adults, beyond the clinical setting.

Based on these premises, I believe the manuscript with minor revisions the article will make a significant contribution to the literature on the role of media and media exposure and its relationship with wellbeing and mental health in young adults.

Here, some minor revisions which I hope will help to strengthen the manuscript even more.

Methodology:

I understood that Item 1 was removed after the EFA due to suboptimal loadings, but its theoretical content is not addressed. I would suggest to briefly discuss what that item measured and why it may have underperformed in this specific cultural or thematic context.

Discussion and Conclusions:

 While the authors cite previous uses of the MVTS, they do not discuss how Italian scores compare to those obtained in other countries (e.g., China).
In conclusion, I suggest adding, within limitations section, also that the samples were mostly drawn from a single Italian region (Campania) and show an overrepresentation of humanities students. Therefore, future studies need to replicate the study across more diverse populations.

In the final section of the manuscript, the usefulness of the scale in individual and group clinical settings is rightly emphasized. However, given that the entire validation process was conducted on a non-clinical population, and that the instrument demonstrated excellent psychometric properties within this group, it would be appropriate to highlight more clearly the potential applicability of the MVTS in non-clinical contexts as well. The scale appears highly suitable particularly among young adults who are heavily exposed to media, even in the absence of clinical symptoms or psychopathological diagnoses. I suggest balancing this section by expanding the discussion on the instrument’s usefulness beyond clinical settings. 

Author Response

Manuscript ID

ejihpe-3805518

Type: Article

Title

Media Exposure and Vicarious Trauma: Italian Adaptation and Validation of the Media Vicarious Traumatization Scale and Its Impact on Young Adults’ Mental Health in Relation to Contemporary Armed Conflicts

Review Report (Reviewer 1)

Comments and Suggestions for Authors

Thank you for the opportunity to revise the manuscript entitled “Media Exposure and Vicarious Trauma: Italian Adaptation and Validation of the Media Vicarious Traumatization Scale and Its Impact on Young Adults’ Mental Health in Relation to Contemporary Armed Conflicts”.

The manuscript presents the Italian adaptation and psychometric validation of the Media Vicarious Traumatization Scale (MVTS), calibrated on the context of contemporary armed conflicts. Across two studies, the authors examine the scale's structure, reliability, validity and measurement invariance across gender. The study includes a total sample of 803 Italian young adults (aged 18–30) and provides evidence for the MVTS’s psychometric soundness.

I believe the study covers a very relevant and timely topic by operationalizing the concept of media vicarious traumatization. The use of both classical test theory and item response theory (IRT) significantly enhances the robustness of the findings. The scale demonstrates strong reliability and validity metrics, and the application of measurement invariance analysis confirms its stability across gender groups. Overall, the MVTS emerges as a useful tool for detecting psychological vulnerability in young adults, beyond the clinical setting.

Based on these premises, I believe the manuscript with minor revisions the article will make a significant contribution to the literature on the role of media and media exposure and its relationship with wellbeing and mental health in young adults.

  • We sincerely thank you for your kind appreciation of our work and for recognizing the adopted methodology as one of its key strengths. We are also grateful for your valuable suggestions, which were greatly appreciated and have contributed to enhancing the quality of our work

Here, some minor revisions which I hope will help to strengthen the manuscript even more.

Methodology:

I understood that Item 1 was removed after the EFA due to suboptimal loadings, but its theoretical content is not addressed. I would suggest to briefly discuss what that item measured and why it may have underperformed in this specific cultural or thematic context.

  • Thank you very much for your suggestion. In paragraph 2.2.2, we have addressed your comment in greater depth by reporting the removal of item 1 and providing, in addition to the statistical justification, a theoretical rationale for its exclusion, namely, the content redundancy with item 2, which had demonstrated a more optimal performance.

Discussion and Conclusions:

 While the authors cite previous uses of the MVTS, they do not discuss how Italian scores compare to those obtained in other countries (e.g., China).      
In conclusion, I suggest adding, within limitations section, also that the samples were mostly drawn from a single Italian region (Campania) and show an overrepresentation of humanities students. Therefore, future studies need to replicate the study across more diverse populations.

  • We have improved sections 1.2 and 4 as requested, specifying that by psychometric properties we refer exclusively to preliminary analyses such as Cronbach’s alpha and the AVE index. Since no prior validation studies of the instrument used have been conducted, a comparison of scores is particularly challenging. However, as reported, in terms of internal consistency and general psychometric properties, our version aligns with recent studies that have employed the MVTS. As suggested, we have included the requested information in the limitations section by adding the following sentence in paragraph 4.1: “Moreover, participants were predominantly recruited from a single Italian region (Campania), and the sample showed an overrepresentation of students from the humanities. Future studies should therefore aim to replicate the research with more diverse populations.

In the final section of the manuscript, the usefulness of the scale in individual and group clinical settings is rightly emphasized. However, given that the entire validation process was conducted on a non-clinical population, and that the instrument demonstrated excellent psychometric properties within this group, it would be appropriate to highlight more clearly the potential applicability of the MVTS in non-clinical contexts as well. The scale appears highly suitable particularly among young adults who are heavily exposed to media, even in the absence of clinical symptoms or psychopathological diagnoses. I suggest balancing this section by expanding the discussion on the instrument’s usefulness beyond clinical settings. 

  • Thank you for this comment. We realized that there was a typo in the term "non-clinical population." As you rightly pointed out, the instrument is validated on a non-clinical population; therefore, we have corrected and expanded the section accordingly. Thank you very much.

Thank you very much for your suggestions; they have significantly improved our paper.

Reviewer 2 Report

Comments and Suggestions for Authors

The research topic is highly relevant and topical, raising very valid questions in the current geopolitical context and reflecting on the impact of war-related media content on young people.
As such, the study presented here (subdivided into two studies) is well constructed and presents data and conclusions that contribute to the advancement of academic knowledge in the field.
However, I believe that this article could be improved in several ways:

1) The title is too long. I think it would be better to reduce it to around 12 words.
2) The abstract does not clearly state the research objectives.
3) Ideally, there should be five keywords. For example, “Future Anxiety” and “Worry about War” do not make sense as keywords.
4) The introduction should not have subheadings. I suggest that these subheadings be grouped together as a literature review.
5) The methodology should be clearer about the research design. The research question and research objectives are missing. Based on these, you should explain and justify the techniques and methods chosen for data collection and processing.

Finally, I must highlight the positive aspects of the final part of the article. In fact, the authors reflected and wrote about the limitations of the research, as well as possible future directions of investigation.

Comments on the Quality of English Language

Overall, the English is easy to read, but a final review is recommended to eliminate typos and sentences that are more complex and less clear.

Author Response

Manuscript ID

ejihpe-3805518

Type: Article

Title

Media Exposure and Vicarious Trauma: Italian Adaptation and Validation of the Media Vicarious Traumatization Scale and Its Impact on Young Adults’ Mental Health in Relation to Contemporary Armed Conflicts

Review Report (Reviewer 2)

Comments and Suggestions for Authors

The research topic is highly relevant and topical, raising very valid questions in the current geopolitical context and reflecting on the impact of war-related media content on young people. 
As such, the study presented here (subdivided into two studies) is well constructed and presents data and conclusions that contribute to the advancement of academic knowledge in the field. 

  • I sincerely thank you for your positive feedback on the topic of our study and the results achieved.

However, I believe that this article could be improved in several ways:

1) The title is too long. I think it would be better to reduce it to around 12 words.

  • I have attempted to revise the title based on your suggestion. However, I believe that this version represents the best effort to clearly and comprehensively convey the themes and objectives of the present work, which, as elaborated in the abstract, go beyond the mere validation of the scale. Nevertheless, I remain available to make further modifications should the reviewers or editors have additional suggestions.

2) The abstract does not clearly state the research objectives.

  • Thank you very much for your suggestion. The abstract has been revised as indicated, and the objectives of the work have been made more explicit.

3) Ideally, there should be five keywords. For example, “Future Anxiety” and “Worry about War” do not make sense as keywords.

  • Thank you for your suggestion; the keywords have been revised and reduced accordingly.

4) The introduction should not have subheadings. I suggest that these subheadings be grouped together as a literature review. 

  • Thank you for your suggestion regarding the structure of the introduction. However, we believe that dividing the literature review into clearly defined subsections improves the organization of the text and facilitates readers’ understanding by focusing on the central themes addressed in this work. This structured approach allows for a clearer and more accessible presentation of the relevant literature.

5) The methodology should be clearer about the research design. The research question and research objectives are missing. Based on these, you should explain and justify the techniques and methods chosen for data collection and processing.

  • Thank you for your suggestion. We have revised paragraph 1.3 to explicitly and clearly present the research question and the research objectives of the two studies. Each "Data Analysis" section of the studies has been tailored to align with the specific objectives described.

Finally, I must highlight the positive aspects of the final part of the article. In fact, the authors reflected and wrote about the limitations of the research, as well as possible future directions of investigation.

  • Thank you very much for your feedback; it is greatly appreciated.

Finally, as suggested, we had the manuscript reviewed once again by a professional translator specialized in academic writing, in order to correct any potential language inaccuracies. Thank you very much for your suggestions; they have significantly improved our work.
